# Describing the Sensory Complexity of Italian Wines: Application of the Rate-All-That-Apply (RATA) Method

**DOI:** 10.3390/foods11162417

**Published:** 2022-08-11

**Authors:** Noemi Sofia Rabitti, Camilla Cattaneo, Marta Appiani, Cristina Proserpio, Monica Laureati

**Affiliations:** Sensory & Consumer Science Lab (SCS_Lab), Department of Food, Environmental and Nutritional Sciences (DeFENS), University of Milan, 20133 Milan, Italy

**Keywords:** rapid descriptive method, RATA, wine quality, red wine, white wine, rosé wine, sparkling wine

## Abstract

The aim of this study was to characterise a large and heterogeneous Italian wine sample applying the Rate-All-That-Apply method (RATA) with semi-trained judges. Twelve judges evaluated 46 samples including white, red, rosé and sparkling wines in two replicates. Judges were asked to select from a list of descriptors all the sensations that described the samples and to evaluate their intensity. Judges obtained high repeatability index scores. A good panel reliability was also highlighted in terms of the reproducibility of the whole sensory characterisation through a multi-factor analysis (MFA). MFA results also showed a good discriminatory ability of the panel with red wines described by bitterness, astringency, body, alcohol and specific olfactory stimuli such as red fruits, spicy and roasted, while white wines were salty, sour and characterised by citrus, tropical fruits and white flowers odours. The RATA method is a suitable and reliable methodology for the description of a wide variety of wine samples and a valuable alternative approach to conventional descriptive analysis to gather information about the sensory perception of a very complex product even when large panels of consumers are not available. Furthermore, the present results provide useful information for wine producers to characterise their products as well as for the optimisation of production disciplinaries, which currently are not exhaustive in the description and the discrimination among products.

## 1. Introduction

In 2021, Italy was confirmed as the world’s leading wine producer with 50.2 million hectolitres produced (19.3% of the total production) [1]. It is well known that the Italian wine scenario is one of the most heterogeneous across time (vintages) and space (regions and specific geographical areas within them), with over 800 wine grape varieties, 20 unique wine regions and thousands of years of wine history [2,3].

In the last decade, Italian wine production has been characterised by an increased quality of the grapes used in winemaking processes, the yeasts applied for the fermentation and the technical procedures of winemakers [4]. However, the concept of “quality” has become much wider nowadays as it results from the interaction of many factors [5] and includes all the properties and intrinsic characteristics of a wine capable of satisfying stated and implicit needs [6]. In this context, sensory analysis is an important tool to discriminate different wines and to evaluate the effect of raw materials and operating conditions adopted during production on products’ final quality [5]. Descriptive sensory analysis is the most powerful tool for quantitatively profiling products and obtaining a complete sensory description [7]. Several descriptive methods have been developed since the 1950s. Quantitative descriptive analysis (QDA) [8] and its subsequent variation as the sensory profiling method [9] have remained the most sophisticated sensory tools to quantitatively profile products [7]. These descriptive methods, referred to as conventional methods, require a highly trained panel of judges to obtain a complete description of a product’s sensory characteristics and their intensity ratings. This implies that describing and quantifying the complex sensory properties of foods and beverages can be an elaborate, time-consuming and expensive task [10,11]. Moreover, the number of samples that can be evaluated at the same time is limited and samples should be more or less homogeneous in their sensory properties. These limitations have led to the development of innovative methodologies for the sensory characterisation of products that require the involvement of consumers or semi-trained judges [11]. The most popular innovative methods proposed to profile a variety of products are Flash Profile [12], Check-All-That-Apply (CATA) [13] and Rate-All-That-Apply (RATA) [14,15]. These methods have been suggested to be rapid, consumer-friendly, flexible and potentially economical in their application [11,16,17].

The CATA method allows product profiles to be quickly obtained, asking consumers to select from a list the attributes that they consider appropriate to describe each product [13,18]. However, an important limitation of this method is the lack of information about selected attributes intensity, resulting in poor discriminability of the samples [11,19]. To overcome this limitation, the RATA method was developed [14,15]. This is an intensity-base variant of CATA where assessors are asked to select all the attributes they consider appropriate to describe the sample and to rate the intensity using a 3-point or a 5-point scale [15]. This methodology can be carried out with consumers [10,19] or semi-trained judges [20]. As shown in several studies, CATA and RATA methods provide similar results in terms of the sensory characterisation of products [14,19,20,21]. However, the RATA method improves the description and discrimination of samples [19,21] as the total number of sensory attributes selected from the judges is usually greater than with CATA [15].

The RATA method has been applied as an alternative descriptive method to profile different types of wines involving large panels of consumers (e.g., [10,22,23,24,25]). However, it should be recognized that, at a company level, it can be challenging to recruit hundreds of people to have an overview of the sensory aspects describing products [26]. This is especially true for winemakers who often belong to very small enterprise realities. Examples of the successful application of the RATA method with a reduced number of trained or semi-trained assessors are available for cream soups [27], milk powders [26] and chocolate [20], but not for wine.

The purpose of the present study was to apply the RATA method to a large and heterogeneous sample of Italian wines, including white, red, rosé and sparkling wines, in order to get a sensory map of some of the most known as well as niche Italian denominations of origin. Our hypothesis is that the RATA method applied with semi-trained assessors is suitable for a reliable sensory characterisation of a wide variety of wines, thus being extremely useful for wine producers and/or small companies to characterise their wine products portfolio even when large panels of consumers are not available.

## 2. Materials and Methods

### 2.1. Participants

A total of 12 assessors (8 women and 4 men) aged between 22 and 44 years (mean age: 30 ± 8 years) were selected from students and employees of the Department of Food, Environmental and Nutritional Sciences of the University of Milan. The number of assessors involved is in line with that of previous studies [20,27]. Only subjects ≥ 18 years of age who were regular wine consumers (wine consumption corresponding to 1–2 glasses per week) were involved in the study. None of the participants had previous or present taste or smell disorders. Informed consent was obtained from all subjects. The study was conducted in accordance with the Declaration of Helsinki and the protocol was approved by the Institutional Ethics Committee (n. 32/12).

### 2.2. Samples

Samples of 46 Italian wines (23 white wines, of which 15 were still and 8 sparkling; 16 red, of which 13 were still and 3 sparkling; and 7 rosés, of which 3 were still and 4 sparkling) were evaluated. All wines were commercial samples, which differed by the area of origin, the vintage, the grape variety and the aging period (Table 1). These wines were chosen because they were part of the portfolio of an eCommerce platform specialised in the sale of Italian wines which are chosen by customers based on their sensory profile characteristics [28].

### 2.3. Vocabulary Development

The first phase of the study consisted of an in-depth analysis of smell, taste and tactile sensations characterising the selected wines. Considering the samples’ heterogeneity in terms of wine type, grape variety and winemaking procedures (e.g., macerated wines, wines aged in barrel), several existing tools (Le Nez du Vin^®^ [29], Tasterplace©, TasterPlace S.r.l., Montebelluna (TV), Italy; The Wine Wheel^®^, Noble et al. [30]) were used as well as literature data [31,32,33,34,35] to cover the sensory complexity of the Italian wines tested, leading to the selection of 169 descriptors: 97 for white wines, 100 for red wines and 133 for sparkling white and rosé wines. Appendix A reports the complete list of descriptors.

The RATA ballot listed the attributes by sensory modality (odour, taste, flavour and tactile sensations). Moreover, in the case of odours and flavours, attributes were listed within macrocategories (e.g., the macrocategory “citrus” included the attributes lemon, orange and grapefruit). This presentation format has been reported to improve attribute processing and reduce cognitive burden in similar tasks [36]. To further ease attribute processing, the order in which the modalities appeared was in line with the expected “dynamics of sensory perception”: (1) odours; (2) flavours; (3) taste; and (4) tactile sensations [20].

### 2.4. Training Phase

The panel underwent six 1 h training sessions according to international guidelines [37,38,39]. During the training period, the judges performed the following tasks related to taste and tactile sensations: (a) taste and tactile sensations identification using both stimuli dissolved in table wine (Appendix A) and water (Appendix A) and (b) ranking of taste stimuli (sweetness, sourness, bitterness) and tactile sensations (astringency and body) in water solutions (Appendix A). Concerning odours and flavours, the judges performed a series of odour identification tasks using cotton buds soaked in pure aromas standards (Le Nez du Vin^®^ [29]; Tasterplace©, TasterPlace S.r.l., Montebelluna (TV), Italy) (Appendix A) and odour/flavour identification tasks using reference standards in table wine (Appendix A).

The training phase also consisted of the familiarisation with the RATA method and the use of structured scales. The training sessions were performed in a collective room and sensory booths at the Sensory & Consumer Science Lab (SCS_Lab) of the Department of Food, Environmental and Nutritional Sciences (DeFENS, Università degli Studi di Milano), designed in accordance with ISO guidelines [40].

### 2.5. Evaluation Phase: The Rate-All-That-Apply (RATA) Method

After the training phase, the 46 wines were evaluated over a 2-month period, with 2 tasting sessions per week, each lasting approximately 1 h. White still (n = 15), red (n = 16) and sparkling white and rosé (n = 8 and n = 7, respectively, for a total of 15) wines were tasted in separate sessions. Sparkling white and rosé wines were evaluated together because they shared several olfactory stimuli. In each session, 5–6 wine samples were evaluated. For each sample, the judges were provided with 15 mL samples served in transparent ISO glasses [41] coded with a 3-digit number and covered with a petri dish to avoid the escape of volatile components. The judges were asked to select all the attributes that described the samples and evaluate their intensity on a 5-point scale (anchors: left, “Low”; middle, “Medium”; right, “High”).

Data acquisition was conducted using Fizz v2.31 software (Biosystèmes, Couternon, France). The judges were asked not to smoke, eat or drink anything, except water, at least one hour before the tasting sessions. The judges evaluated the samples under white light and were provided with mineral water and unsalted crackers to clean their mouth between tastings. Wines were served at room temperature, which was set at 20 °C. Each wine sample was evaluated in duplicate under the same experimental conditions. Presentation orders were systematically varied over the judges and replicates to balance the effects of serving order and carryover [42].

### 2.6. Data Analysis

First, the reliability of the panel in selecting the descriptors in the two replicates was checked by calculating, for each assessor and for each wine, a reliability index (RI) according to Giacalone and Hedelund [20]:$$RI_{j}=\frac{1}{n}*\overset{n}{\underset{s=1}{\sum }}\left(\frac{des_{com\;s}}{des}\right)$$
where *RI_j_* is the global reliability index related to judge *j*, *n* is the number of samples (n = 15 for white wines, 16 for red wines and 15 for sparkling white and rosé wines), *des_com s_* is the number of descriptors indicated by judge *j* in an identical manner in all the replicates for a given sample *s* (including the descriptors checked as well as the ones not checked by the judge in both replicates) and *des* is the total number of descriptors including all odours, flavours, taste and tactile sensations descriptors (n = 97 for white wines, n = 100 for red wines and n = 133 for sparkling white and rosé wines).

The index was calculated separately for white, red and sparkling white and rosé wines. RI can assume values from 0 to 1; the higher the RI value, the higher the reliability of the judge. An RI value ≥ 0.5 was considered as the cut-off for the judges’ reliability according to Giacalone and Hedelund [20].

In order to account for the limitation highlighted by Worch and Piqueras-Fiszman [43] that the RI might be too optimistic if the experimental conditions inflate the number of unchecked items, a further calculation was performed in which the descriptors that were selected less than 10% of the time by the judges were omitted. This led to the omission of 16, 8 and 13 descriptors and a very limited reduction of RIs corresponding to 4%, 2% and 2% for white, red and sparkling white and rosé wines, respectively.

The judges’ reliability was also evaluated in terms of repeatability in rating the intensity of the descriptors for each wine through a multiple factor analysis (MFA) run on intensity data provided by the judges in both replicates [20].

Once the judges’ reliability was checked, MFAs were performed on rating data averaged across judges and replicates for the three separate wines categories (white, red and sparkling white and rosé wines) in order to provide a detailed sensory characterisation of each wine type. All attributes that loaded less than ±0.25 were excluded from the analysis [44]. The XLSTAT software (Addinsoft, Boston, MA, USA) was used to perform MFAs.

## 3. Results

### 3.1. Panel Reliability

RIs calculated by judge and averaged across white, red and sparkling white and rosé wines are reported in Appendix A. All RI values were well above the cut-off value of 0.5 reported by Giacalone and Hedelund [20], indicating the good reliability of the panel in selecting the same attributes in the two replicates.

Panel reliability was further assessed in terms of the reproducibility of the whole sensory characterisation by evaluating the configurational similarity of product spaces obtained from the two separate replicates (Appendix A). This analysis is reported to show the proximity of the replicates for each wine and the ability of the judges to discriminate between wine categories as evidence of the good panel reliability.

### 3.2. Wines Sensory Characterisation

#### 3.2.1. White Wines

MFA results of the 15 white wines are reported in Figure 1a,b. The first two dimensions explained 42.98% of the total variance. In the upper left quadrant of Figure 1b, the macrocategories of caramelised (caramel, honey, butter), spices (vanilla), dried fruit (fig and raisins), nuts (almond, walnut and hazelnut), woody (cedar wood), floral (chamomile, acacia) and balsamic (eucalyptus and thyme) were perceived in Nosiola wine (NOS-PAL), which underwent a maceration process.

In the lower left quadrant, the fruity macrocategories, including tree fruit (apple, pear, peach and apricot), tropical (pineapple, melon, and lychee) and citrus (orange, lemon and grapefruit), are located corresponding to wines such as Collio Ribolla (COL), Verdicchio (VRD), Lugana (LUG), and Nosiola (NOS_VIG). The wines Fiano (FIA), Gambellara Classico (GAM) and Vermentino (VRM) share the aforementioned odour macrocategories of the left quadrants. Conversely, these wines were those that obtained the lowest scores for the odour macrocategories earthy (mushroom, truffle, lees and musk) and ethereal (fuel), as well as the odour and flavour of saffron and toasted bread, which are placed in the upper right quadrant of Figure 1b, corresponding to Coda di Volpe wine (COD) and to a lesser extent to Biancolella (BIA), Pecorino (PEC) and Collio Malvasia (MAL). In the lower right quadrant of Figure 1b, the odour macrocategories of floral (jasmine and linden), ethereal (solvent) and tropical fruit (pineapple, melon and banana) describing the Kerner (KER), Pignoletto (PIG) and Albana (ALB) wines are mainly located. The taste and tactile sensations are located in the right panes. Pignoletto (PIG) and Albana (ALB) wines had the highest alcohol content (13.5 and 14.5% vol, respectively,) and, accordingly, they were perceived with the highest intensity of alcohol as well as sourness. Coda di Volpe wine (COD) was perceived with a high intensity of salty taste and body.

#### 3.2.2. Red Wines

MFA results of the 16 red wines are reported in Figure 2a,b. The first two dimensions explained 54.1% of the total variance. In the right quadrants of Figure 2b, the odour macrocategories of red fruits (black sour cherry, cherry, pomegranate, strawberry), wild berries (blackberry, blueberry, raspberry, currant), floral (orange blossom, dried flowers, rose, violet) and baked fruit (fruit jam and dried plum) are located. These descriptors are mainly driven by Sangue di Giuda wine (SAN) but contribute also to the characterisation of Colli Berici Tai Rosso (TAI_RED), Bonarda Oltrepò Pavese (BON), Nero d’Avola (NER_AVO) and Lacrima di Morro d’Alba (LAC_RED) wines. The upper left quadrant of Figure 2b is mainly characterised by the odour macrocategories of spices (liquorice, pepper, cloves, vanilla), and roasted (coffee, chocolate), which mainly characterise Avanà (AVA_RED), Rossese Dolceacqua (ROS), Riviera del Garda Classico (RIV) and Montepulciano (MON) wines. Benaco bresciano (BEN) and Buttafuoco (BUT) wines were mainly perceived in terms of nuts (walnut, almond, hazelnut) and wine body. Finally, in the lower left quadrant of Figure 2b are positioned the odour macrocategories of balsamic (thyme, eucalyptus, mint, anise), vegetative (hay, oregano, sage, bell pepper), roasted (smoke, tobacco, toasted bread) as well as specific olfactory stimuli such as ethereal, leather and oak, attributable to Rubicone Centesimino (RUB), Chianti (CHI), Ciliegiolo (CIL) and Nero di Troia (NER_TRO) wines.

#### 3.2.3. Rosé and White Sparkling Wines

MFA results of the seven rosé wines and eight sparkling white wines are reported in Figure 3a,b. The first two dimensions explained 53.21% of the total variance. Comparing Figure 3a,b, Recioto Spumante Metodo Classico (REC), positioned in the upper right quadrant (Figure 3a), was clearly distinguished by all wines being characterised mainly by caramel odour and the macrocategories of nuts (almond, hazelnut), dried fruits (figs, prune, raisins), ethereal (flint stone, solvent) and vegetative (rosemary, sage, marjoram). The two Valdobbiadene DOCG (VAL_BRU and VAL_EXD) and Moscato d’Asti (MOS) wines, located in the lower right quadrant, were also clearly separated from all other wines and were mainly characterised by the macrocategories of floral (linden, hawthorn, acacia, chamomile, jasmine, orange blossom), caramelised (honey, vanilla) and fruit including tropical (pineapple, litchi, melon), tree fruit (apple, pear, quince, Moscato grape, peach, apricot) and baked/ripe fruit. All the wines located in the right panes were perceived as sweet and with a high wine body, while those in the left panes were sour, salty and astringent. Those wines are all the rosé wines and some white sparkling wines, including Alta Langa DOCG Extra Brut (ALT_WHI), Trento Metodo Classico DOCG (TRE), Franciacorta DOCG brut (FRA) and Marche IGT (MAR). Concerning olfactory stimuli, in the left quadrants the macrocategories of red fruits, red flowers and wild berries are located, which were driven by rosé wines, as well as the macrocategories of citrus (grapefruit, orange), earthy (mushroom, lees), vegetative (fresh-cut grass, marjoram, sage) and yeast (yeast, bread crust), which distinguished Alta Langa DOCG Extra Brut (ALT_WHI), Trento Metodo Classico DOCG (TRE), Franciacorta DOCG brut (FRA) and Marche IGT (MAR) wines.

## 4. Discussion

In this study, a sensory description of a large and heterogeneous sample of Italian wines characterised by different vintages, origins and oenological techniques was obtained by applying the RATA method involving semi-trained judges. In the present study, the results showed that the panellists had an adequate performance as a whole and individually. Moreover, the wine sensory maps obtained through MFAs showed a good discrimination ability of the panel, indicating the suitability of the RATA method to characterise the sensory properties of very different wines. Ares et al. [15] highlighted that the efficacy of RATA questions may depend on the specific product category being tested. For instance, conventional descriptive methods using attribute intensity measures may be more relevant for simple products (where differences might be small), while applicability measures such as CATA and RATA may be more appropriate for sensory characterisations of complex and heterogeneous products, such as wine. Moreover, it has been reported that the RATA method works well when a large consumer panel is involved [16,21]. Although in this study a panel of consumers could have been involved, it should be acknowledged that a high number of people is not always easy to recruit, especially in R&D situations or more generally at the company level. Moreover, wine is an exceptionally complex product from a sensory point of view, and considering that the samples evaluated in the present study were extremely heterogeneous, with different wine styles (white, red, rosé and sparkling) as well as different denominations of origin within the same wine style, the number of descriptors used in the ballot (n = 97 for white wines, n = 100 for red wines and n = 133 for rosé and sparkling wines) was much higher than any other study of which we are aware (min number of descriptors n = 10 in Sinesio et al. [46], max number of descriptors n = 58 in Mezei et al. [23]). Such a wide list of descriptors would hardly have been feasible to manage by consumers.

To the best of our knowledge, this is the first paper that applied the RATA method to characterise so wide a range of wine products with semi-trained assessors.

Despite it being largely recognized that intrinsic tasting experience is the most important reason for drinking wine [47], surprisingly enough, there is a paucity of studies related to the description of the sensory properties of this product. A recent review on the trends in the oenological and viticulture sectors [48] highlighted that, among the most studied topics in the field, sensory analysis only sporadically emerged. Studies that describe wine quality using sensory methodologies are mainly focused on the characterisation of different vintages from the same grape variety (e.g., [49]), different varieties from the same producing area (e.g., [50]) or different producing areas for the same grape variety (e.g., [51]).

With regard to white wines, Nosiola (NOS_PAL) was consistently evaluated as rich in hazelnut and caramelized notes, with hints of vanilla and dried fruits [52]. These descriptors are typical of wines that have undergone a maceration process on grapes [53]. Judges evaluated the Vermentino sample (VRM) as floral, citrus and spicy with hints of tropical and dried fruit, results that are confirmed in an earlier study by Asproudi et al. [54]. Furthermore, in agreement with Robinson et al. [55], Vermentino seems to have hints of nuts. According to previous data, the Fiano sample (FIA) has been characterised by fruit tree, honey, floral, exotic fruit and citrus aromas [56,57,58]. Malvasia (MAL) has been defined as a mineral/ethereal wine characterised by an earthy and toasted bread aroma, whereas it was previously described as a fruity and floral wine [59,60,61]. This discrepancy is probably due to the different terroir and production regions considered in the aforementioned papers, which did not allow further comparisons. Coda di Volpe (COD) was correctly evaluated as one of the samples with the highest intensity of wine body [62]. Gambellara Classico (GAM), Collio Ribolla (COL), Verdicchio (VRD), Lugana (LUG) and Nosiola (NOS_VIG) wines revealed a clear character of tree fruit (apple, pear, peach and apricot), tropical (pineapple, melon, and lychee) and citrus (orange, lemon and grapefruit) aromas. The wines Biancolella (BIA) and Pecorino (PEC) were described as exhibiting the odour macrocategories of earthy (mushroom, truffle, lees and musk) and ethereal (fuel), as well as the odour and flavour of saffron and a toasted bread odour. To the best of our knowledge no literature data on sensory description are available on the previously reported wines. Thus, a comparison could not be provided.

Concerning red wines, Chianti (CHI), Ciliegiolo (CIL) and Nero di Troia (NER_TRO) samples were described by balsamic, vegetative and roasted macrocategories as well as by ethereal, leather and oak aromas. Literature data revealed that the two last-mentioned red wines are generally described by floral and fruity macrocategories [63,64], while Chianti has been associated with woody and roasted aromas [65]. Considering tactile stimuli, Chianti (CHI) was correctly evaluated as one of the samples with the highest astringency [66]. According to Ubigli and Cravero [35], Sangue di Giuda (SAN) was mainly characterised by red and wild fruits, while Montepulciano (MON) was spicy [55].

The literature concerning sparkling wines is mainly related to Valdobbiadene (VAL_BRU; VAL_EXD) [67] and Recioto Spumante Metodo Classico (REC) wines [68] and is consistent with the results obtained in the present study. Recioto Spumante Metodo Classico (REC), a typical Italian sweet wine, was mainly characterised by a caramel odour and the macrocategories of nuts and dried fruits. These aromas are probably the results of the drying of the grapes [31], which are left for a couple of months on wooden trellises to further dehydrate and increase their sugar content, aromas and noble rots [69]. The two Valdobbiadene DOCG wines (VAL_BRU and VAL_EXD) were mainly characterised by the macrocategories of floral (linden, hawthorn, acacia, chamomile, jasmine, orange blossom), caramelised (honey, vanilla) and fruit including tropical (pineapple, litchi, melon), tree fruit (apple, pear, quince, Moscato grape, peach, apricot) and baked/ripe fruit, in accordance with previous data [70]. As well, floral and fruity descriptors have been associated with Moscato d’Asti (MOS), which has been characterised as having a “sweet” taste, in accordance with the high residual sugar and low alcohol content in wine dessert [35].

To the best of our knowledge, very limited sensory research has been conducted on rosé wines; therefore, the results obtained in the present study could be useful to compensate for this literature gap. Rosé wines were described by the presence of aromas and flavours of red fruits (for example, strawberry and cherry) [71] typical of red wines as well as olfactory stimuli more representative of white wines, e.g., citrus. Still and sparkling rosé wines were also perceived with a higher bitterness and astringency than white sparkling wines, which is coherent with the wine-making process characterising rosé wines [72], as well as with the low wine body and alcohol. Accordingly, Ribéreau-Gayon et al. [72] described rosé wines as being fruity and with a light structure, while Jackson [73] reported bitterness as one of the main features of rosé wines.

Some limitations of the study should be acknowledged. The characterisation of the wines considered only one producer by wine type, thus lacking representativeness. The effectiveness of the method in providing an exhaustive characterisation and satisfactory product discrimination has not been verified in comparison with other methods. Furthermore, the models obtained by MFA returned somewhat low variance values, which could be ascribed to the fact that the judges were semi-trained and not trained. Future perspectives of study might be to test the RATA method involving semi-trained judges in combination with conventional descriptive approaches to verify whether the discrimination and descriptive capacity of the panel may depend on the complexity of the product.

## 5. Conclusions

The RATA method has proven to be a suitable and reliable methodology for the description of a wide variety of wine samples. In this sense, the RATA method could be a simple and valuable alternative approach to conventional descriptive analysis to gather information about the sensory perception of a very complex product. Apart from the methodological contribution, this study also offers an overview of the sensory complexity of some Italian wines, providing useful information for wine producers to characterise their products, thus contributing to improving marketing strategies. Furthermore, the results obtained could be useful for the optimisation of the production disciplinaries which currently do not provide an exhaustive sensory description and do not allow discrimination among products.

## Figures and Tables

**Figure 1 foods-11-02417-f001:**
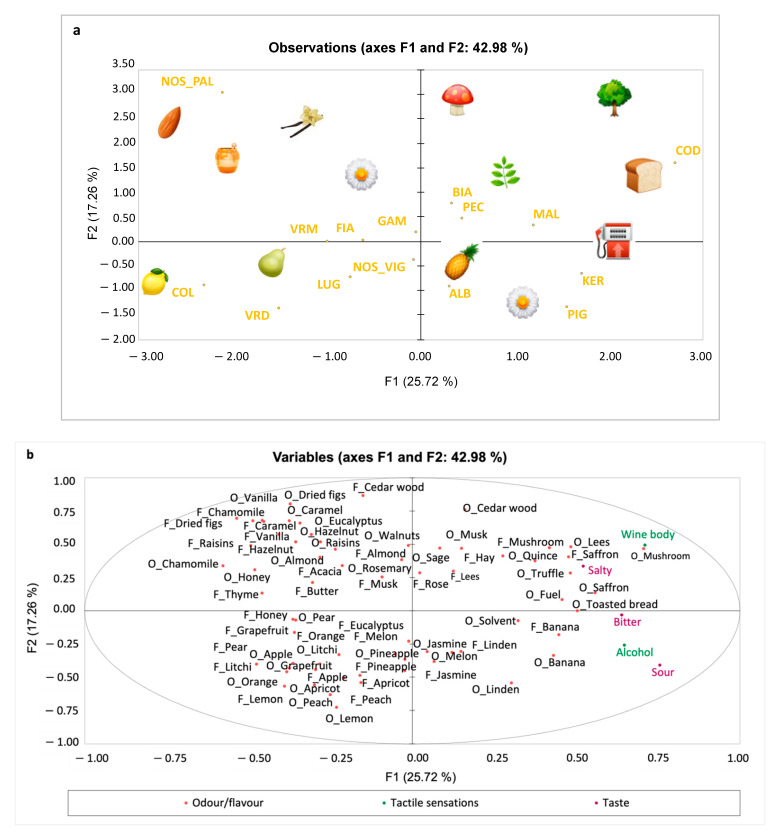
Samples (**a**) and variables (**b**) configuration of MFA performed on the 15 white wines (data averaged across replicates). Variables excluded as they loaded less than ± 0.25 [44]: F_Quince, F_Walnuts, O_Rose, O_Acacia, O and F_Hawthorn, O_Hay, F_Sage, F_Rosemary, O_Thyme, O and F_Mint, F_Truffle, F_Toasted Bread, O and F_Pine, O_Butter, O and F_Candied citron, O and F_Flint Stone, F_Solvent and F_Fuel. The emoji shown in Figure 1a are from the site: https://emojipedia.org/apple-watch/ (accessed on 5 July 2022) [45]. ALB—Romagna Albana DOCG; BIA—Ischia DOC Biancolella; COD—Coda di Volpe DOC; COL—Collio Ribolla Gialla DOC; FIA—Fiano di Avellino DOCG; GAM—Gambellara Classico DOC; KER—Vigneti delle Dolomiti IGT, Kerner; LUG—Lugana Riserva DOC; MAL—Collio Malvasia DOC; NOS_PAL—Nosiola Palustella Trentino DOC; NOS_VIG—Vigneti delle Dolomiti IGT, Nosiola; PEC—Pecorino DOP; PIG—Colli Bolognesi Pignoletto Superiore DOCG; VRD—Verdicchio Dei Castelli Di Jesi DOC Classico Superiore; VRM—Vermentino DOC.

**Figure 2 foods-11-02417-f002:**
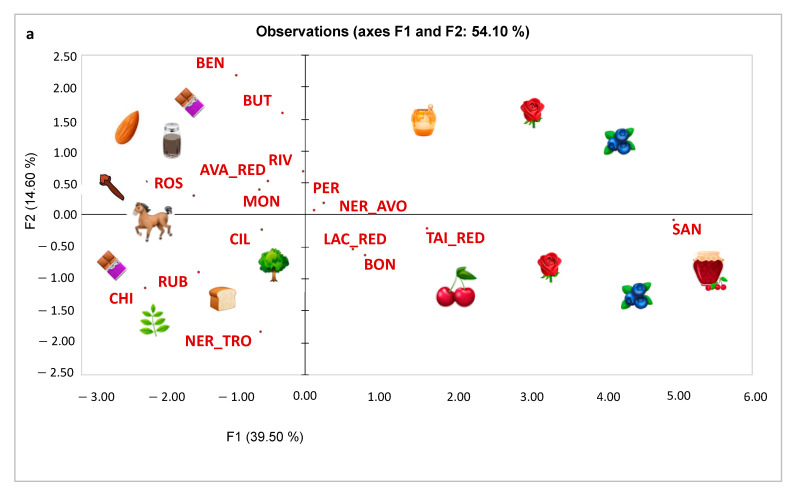
Samples (**a**) and variables (**b**) configuration of MFA performed on the 16 red wines (data averaged across replicates) and (b) relevant sensory descriptors configuration. Variables excluded as they loaded less than ± 0.25 [44]: O_Cherry, O_Mulberry, O_Almond, F_Walnuts, O_Dried flowers, O_Sage, O_Hay, O and F_Fresh cut grass, O_Rosemary, O_Oregano, O and F_Anise, O_Vanilla, O and F_Cinnamon, O and F_Mushroom, O and F_Musk, O and F_Truffle, O_Lees, F_Toasted bread and F_Butter. The emoji shown in Figure 2a are from the site: https://emojipedia.org/apple-watch/ (accessed on 5 July 2022) [45]. AVA_RED—Valsusa DOC; BEN—Benaco Bresciano IGT; BON—Bonarda dell’Oltrepò Pavese DOC; BUT—Buttafuoco dell’Oltrepò Pavese DOC; CHI—Chianti Superiore DOCG; CIL—Maremma Toscana DOC; LAC_RED—Lacrima di Morro d’Alba DOC; MON—Colline Teremane Montepulciano d’Abruzzo DOCG; NER_AVO—Nero d’Avola Menfi DOC; NER_TRO—Cacc’e Mmitte di Lucera DOC; PER—Perricone Terre siciliane IGT; RIV—Riviera del Garda Classico DOC; ROS—Rossese Di Dolceacqua Superiore DOC; RUB—Rubicone Centesimino IGT; SAN—Sangue di Giuda dell’Oltrepò Pavese DOC; TAI_RED—Colli Berici Tai Rosso DOC.

**Figure 3 foods-11-02417-f003:**
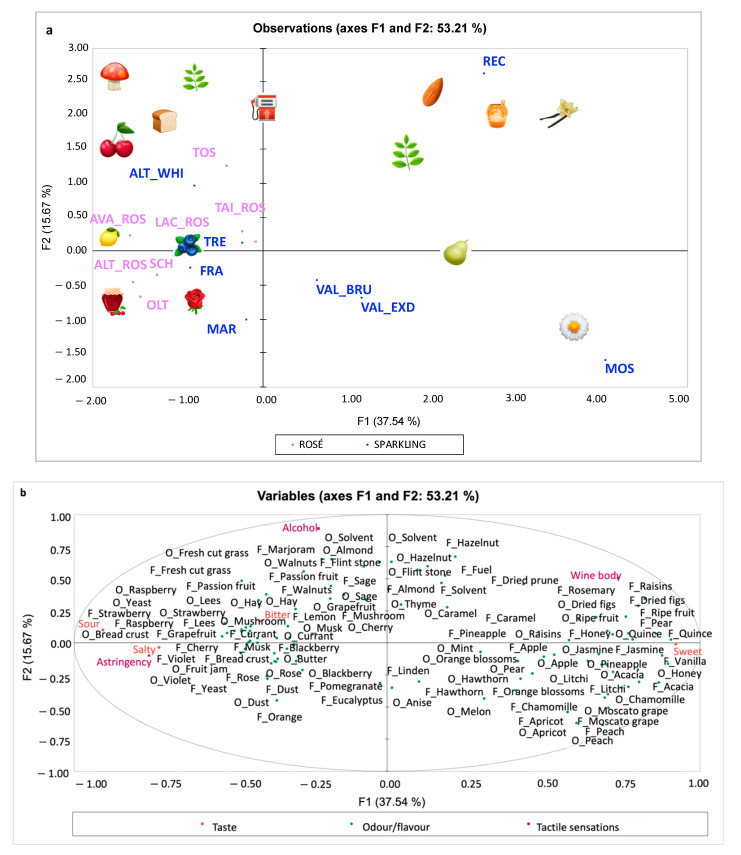
Samples (**a**) and variables (**b**) configuration of MFA performed on 7 rosé and 8 sparkling white wine samples, odour/flavour (107 variables), tactile sensations (3 variables) and taste (4 variables). Variables excluded as they loaded less than ± 0.25 [44]: O_Lemon; O_Orange; O and F_Banana; F_Melon; O_Passion fruit; O_Dried prune; F_Fruit jam; O_Linden; O and F_Dried flowers; O_Rosemary; F_Thyme; F_Mint; O_Eucalyptus; F_Anise; O and F_Saffron; O and F_Truffle; O and F_Toasted bread; O and F_Oak; F_Butter; and O_Fuel. The emoji shown in Figure 3a are from the site: https://emojipedia.org/apple-watch/ (accessed on 5 July 2022) [45]. *SPARKLING WINES*: ALT_WHI—Alta Langa DOCG Extra Brut; FRA—Franciacorta DOCG Brut; MAR—Marche IGT; MOS—Moscato d’Asti DOCG; REC—Recioto Spumante Metodo Classico DOCG; TRE—Trento Metodo Classico DOCG millesimato; VAL_BRU—Valdobbiadene DOCG Brut; VAL_EXD—Valdobbiadene DOCG Extra Dry. *ROSÉ WINES*: ALT_ROS—Alta Langa DOCG; AVA_ROS—Vino Rosato Frizzante; LAC_ROS—Spumante Rosato Brut; OLT—Oltrepò Pavese Metodo Classico Pinot Nero Rosé DOCG; SCH—Vigneti delle Dolomiti IGT, Schiava; TAI_ROS—Colli Berici Tai Rosato DOC; TOS—Toscana IGT.

**Table 1 foods-11-02417-t001:** Wine samples’ characteristics.

Label	Type	Wine	Grape Variety	Area of Production	Vintage	Vol.%
ALB	White	Romagna Albana DOCG	100% Albana	Forlì-Cesena, Emilia-Romagna, Centre of Italy	2018	14.5 vol.%
BIA	White	Ischia DOC Biancolella	100% Biancolella	Ischia Island, Campania, Southern Italy	2018	13 vol.%
COD	White	Coda di Volpe DOC	100% Coda di Volpe	Avellino, Campania, Southern Italy	2019	13 vol.%
FIA	White	Fiano di Avellino DOCG	100% Fiano	Avellino, Campania, Southern Italy	2019	13 vol.%
GAM	White	Gambellara Classico DOC	100% Garganega	Vicenza, Veneto, Northern Italy	2018	13 vol.%
LUG	White	Lugana Riserva DOC	100% Turbiana	Brescia, Lombardia, Northern Italy	2017	14 vol.%
KER	White	Vigneti delle Dolomiti IGT, Kerner	100% Kerner	Trento, Trentino-Alto Adige, Northern Italy	2019	13.5 vol.%
NOS_PAL	White	Nosiola Palustella Trentino DOC (Organic)	100% Nosiola Trentina	Trento, Trentino-Alto Adige, Northern Italy	2019	12.5 vol.%
NOS_VIG	White	Vigneti delle Dolomiti IGT, Nosiola (Organic)	100% Nosiola	Trento, Trentino-Alto Adige, Northern Italy	2019	12.5 vol.%
PIG	White	Colli Bolognesi Pignoletto Superiore DOCG	100% Grechetto gentile	Bologna, Emilia-Romagna, Centre of Italy	2019	13.5 vol.%
COL	White	Collio Ribolla Gialla DOC	100% Ribolla Gialla	Gorizia, Friuli-Venezia Giulia, Northern Italy	2019	12.5 vol.%
VRM	White	Vermentino DOC	100% Vermentino	Oristano, Sardegna, Centre of Italy	2018	13.5 vol.%
VRD	White	Verdicchio Dei Castelli Di Jesi DOC Classico Superiore (Organic)	100% Verdicchio	Ancona, Marche, Centre of Italy	2019	12.5 vol.%
PEC	White	Pecorino DOP (Organic)	100% Pecorino	Teramo, Abruzzo, Centre of Italy	2018	12.5 vol.%
MAL	White	Collio Malvasia DOC	100% Malvasia	Gorizia, Friuli-Venezia Giulia, Northern Italy	2019	13 vol.%
RUB	Red	Rubicone Centesimino IGT(Organic)	100% Centesimino	Forlì-Cesena, Emilia-Romagna, Centre of Italy	2019	15 vol.%
RIV	Red	Riviera del Garda Classico DOC	85% Groppello, 15% Marzemino, Sangiovese and Barbera	Brescia, Lombardia, Northern Italy	2019	13.5 vol.%
BEN	Red	Benaco Bresciano IGT	70% Rebo, 15% Cabernet Sauvignon, 15% Marzemino appassito	Brescia, Lombardia, Northern Italy	2017	14.5 vol.%
MON	Red	Colline Teramane Montepulciano d’Abruzzo DOCG	100% Montepulciano d’Abruzzo	Teramo, Abruzzo, Centre of Italy	2018	13.5 vol.%
PER	Red	Perricone Terre Siciliane IGT (Organic)	100% Perricone	Agrigento, Sicilia, Southern Italy	2020	12 vol.%
NER_AVO	Red	Nero d’Avola Menfi DOC (Organic)	100% Nero d’Avola	Agrigento, Sicilia, Southern Italy	2020	12.5 vol.%
ROS	Red	Rossese Di Dolceacqua Superiore DOC	97% Rossese di Ventimiglia, 3% red non-aromatic grapes	Imperia, Liguria, Northern Italy	2019	13.5 vol.%
LAC_RED	Red	Lacrima di Morro d’Alba DOC (Organic)	100% Lacrima	Ancona, Marche, Centre of Italy	2020	13 vol.%
BON	Red sparkling	Bonarda dell’Oltrepò Pavese DOC	100% Croatina	Pavia, Lombardia, Northern Italy	2020	13.5 vol.%
BUT	Red sparkling	Buttafuoco dell’Oltrepò Pavese DOC	Croatina, Barbera and Ughetta di Solinga (variable % depending on vintages)	Pavia, Lombardia, Northern Italy	2019	13.5 vol.%
CHI	Red	Chianti Superiore DOCG	90% Sangiovese, 10% Ciliegiolo	Pisa, Toscana, Centre of Italy	2017	14 vol.%
AVA_RED	Red	Valsusa DOC	100% Avanà	Torino, Piemonte, Northern Italy	2020	14 vol.%
NER_TRO	Red	Cacc’e Mmitte Di Lucera DOC	60% Nero di Troia, 30% Montepulciano, 10% Bombino	Foggia, Puglia, Southern Italy	2019	13 vol.%
SAN	Red sparkling	Sangue di Giuda dell’Oltrepò Pavese DOC	40% Croatina, 40% Barbera, 20% Uva rara	Pavia, Lombardia, Northern Italy	2020	6 vol.%
CIL	Red	Maremma Toscana DOC	100% Ciliegiolo	Grosseto, Toscana, Centre of Italy	2019	13.5 vol.%
TAI_RED	Red	Colli Berici Tai Rosso DOC	100% Tai Rosso	Vicenza, Veneto, Northern Italy	2019	12 vol.%
TRE	White Sparkling	Trento Metodo Classico DOCG Millesimato	100% Chardonnay	Trento, Trentino-Alto Adige, Northern Italy	2017	12.5 vol.%
VAL_BRU	White Sparkling	Valdobbiadene DOCG Brut	100% Glera	Treviso, Veneto, Northern Italy	2019	12 vol.%
VAL_EXD	White Sparkling	Valdobbiadene DOCG Extra Dry	100% Glera	Treviso, Veneto, Northern Italy	2019	12 vol.%
FRA	White Sparkling	Franciacorta DOCG Brut	75% Chardonnay, 20% Pinot nero5% Pinot bianco	Brescia, Lombardia, Northern Italy	2019	12.5 vol.%
ALT_WHI	White Sparkling	Alta Langa DOCG Extra Brut	60% Chardonnay, 40% Pinot nero	Cuneo, Piemonte, Northern Italy	2017	12.5 vol.%
MAR	White Sparkling	Marche IGT	80% Verdicchio, 20% Trebbiano	Ancona, Marche, Centre of Italy	2020	11 vol.%
REC	White Sparkling	Recioto Spumante Metodo Classico DOCG	100% Garganega	Vicenza, Veneto, Northern Italy	2017	13 vol.%
MOS	White Sparkling	Moscato d’Asti DOCG	100% Moscato bianco	Cuneo, Piemonte, Northern Italy	2020	5.5 vol.%
SCH	Rosé	Vigneti delle Dolomiti IGT, Schiava	100% Schiava	Trento, Trentino-Alto Adige, Northern Italy	2019	11.5 vol.%
ALT_ROS	Rosé Sparkling	Alta Langa DOCG	100% Pinot nero	Cuneo, Piemonte, Northern Italy	2017	12.5 vol.%
LAC_ROS	Rosé Sparkling	Spumante Rosé Brut	100% Lacrima	Ancona, Marche, Centre of Italy	2020	12.5 vol.%
AVA_ROS	Rosé Sparkling	Sparkling Rosé wine	100% Avanà	Torino, Piemonte, Northern Italy	2018	12 vol.%
OLT	Rosé Sparkling	Oltrepò Pavese Metodo Classico Pinot Nero Rosé DOCG	100% Pinot noir	Pavia, Lombardia, Northern Italy	2018	12.5 vol.%
TAI_ROS	Rosé	Colli Berici Tai Rosato DOC	100% Tai Rosso	Vicenza, Veneto, Northern Italy	2020	12 vol.%
TOS	Rosé	Toscana IGT (Organic)	100% Sangiovese	Pisa, Toscana, Centre of Italy	2020	13 vol.%

## Data Availability

Data will be made available by the corresponding author upon reasonable request.

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
