# Peer review of "Describing the Sensory Complexity of Italian Wines: Application of the Rate-All-That-Apply (RATA) Method"

_foods, 2022, doi:10.3390/foods11162417_

Round 1

Reviewer 1 Report

Rate-All-That-Apply ( RATA)  is a suitable method to evaluate the wine sensory quality and profiling. However, the accuracy should be improved if replace the QDA methods. In this manuscript, lots of work have been done for sensory profile of Italian wines.

Line 30, In 2020, Italy has been confirmed as the world's leading wine producer with 49 30 million hectolitres produced (19 % of the total production) [1].

Wine production in major countries in 2021 have been released by OIV, so, it is better to use the latest data.

Line34, only hundreds of years of wine history?

Line 77-79, Examples of the successful application of the RATA method with a 77 reduced number of trained or semi-trained assessors are available for cream soups [26], 78 milk powders [25] and chocolate [19], but not for wine.

RATA have been used for evaluation of sensory profiling of wine(Based on the Web of Science)

LINE 231-232, MFA results of the 15 white wines are reported in Figure 1a-b. The first two dimensions explained 42.98 % of the total variance. It is too low to explain the total variance (42.98 %). Similarly, for red wine and sparkling wine. So, we should reconsider this method is reliable or not, and why?

Author Response

REVIEWER 1:

Rate-All-That-Apply ( RATA)  is a suitable method to evaluate the wine sensory quality and profiling. However, the accuracy should be improved if replace the QDA methods. In this manuscript, lots of work have been done for sensory profile of Italian wines.

We sincerely thank the reviewer for the time dedicated to evaluate our manuscript

Line 30, In 2020, Italy has been confirmed as the world's leading wine producer with 49 30 million hectolitres produced (19 % of the total production) [1].

Wine production in major countries in 2021 have been released by OIV, so, it is better to use the latest data. 

Dear reviewer, thank you for the suggestion, the authors have made the update.

Line34, only hundreds of years of wine history?

The authors are aware that the Italian wine history is very ancient. By “hundreds” we do not mean 100 years but several hundred years. The sentence has been revised  in the manuscript to avoid misleading information

Line 77-79, Examples of the successful application of the RATA method with a 77 reduced number of trained or semi-trained assessors are available for cream soups [26], 78 milk powders [25] and chocolate [19], but not for wine.

RATA have been used for evaluation of sensory profiling of wine (Based on the Web of Science)

The authors are aware that there are many articles that use RATA with wine, however these studies involve a large number of consumers and not trained or semi-trained judges as reported at in lines 72 and 73. As outlined in the manuscript, in the field of wine science, it can be difficult to find large numbers of consumers because wine producers very often belong to small enterprise realities. The present study refers to the RATA method used with semi-trained judges, for which, to the best of our knowledge, there are no examples in the literature concerning wine. For this reason, the authors believe the manuscript can be of interest for practitioners in the field. A list of articles using the RATA method with wine involving a large number of inexpert judges is reported below. In the present manuscript, only some of these many articles have been cited, however if the reviewer think that it can be helpful to improve the quality of the manuscript (or if he/she has other articles of which we are unaware), the authors may cite other research done on this topic.

Authors

year

Doi

type panel

n. panel

Bottcher et al

2022

doi: 10.1111/ajgw.12516

Consumers

60

Wilkinson et al

2022

doi: 10.1111/ajgw.12548

consumers

51

Wang et al

2022

https://doi.org/10.1016/j.foodchem.2021.131222

consumers

61

Sinesio et al

2021

https://doi.org/10.1016/j.foodqual.2021.104268

consumers

108

Gonzaga et al

2021

DOI:10.20870/oeno-one.2021.55.4.4805

consumers

60

Hranilovic et al

2021

https://doi.org/10.1016/j.foodchem.2021.129015

wine experts

47

Mezei et al

2021

DOI:10.20870/oeno-one.2021.55.2.4571

wine experts

43

Armstrong et al

2021

doi: 10.1111/ajgw.12469

consumers

59

Golombek et al

2021

https://doi.org/10.1016/j.foodchem.2020.128003

inexperts

21

Gonzaga et al

2022

https://doi.org/10.1016/j.foodres.2021.110719

consumers

112

Sáenz-Navajas et al

2017

http://dx.doi.org/10.1016/j.foodres.2017.02.002

wine experts

39

Nguyen et al

2020

doi:10.3390/foods9020224

consumers

69

LINE 231-232, MFA results of the 15 white wines are reported in Figure 1a-b. The first two dimensions explained 42.98 % of the total variance. It is too low to explain the total variance (42.98 %). Similarly, for red wine and sparkling wine. So, we should reconsider this method is reliable or not, and why?

Thank you for your comment, which the authors share. Actually, when a sensory profiling method with trained judges is applied, the variance returned from multivariate analysis is much higher, in the present case the authors assume that the lower variance is due to the fact that the judges were semi-trained. This point is quoted in the discussion section of the revised version of the manuscript.

Reviewer 2 Report

The application of RATA method in wine analysis is not new but as the authors say never to such large and heterogeneous sample of wines. The main problem of this work is the presentation of achieved results. Firstly explain was the list of modalities for odour/flavour one or there were two separate list, one for odour and one for flavour...also in the figure there are marks for odour -O, flavour -F but also O/F...please explain how this was defined. All the figures  with variables (figure b) have too much data and are not easy to read. My suggestion is to present the data in separate figures, one dealing with odours, one with flavours and the one with taste and tactile sensation. In this way also much better discussion of results can be performed.

Section 2.5 is written twice..

Please explain why the wines were tasted at room temperature ( 20 °C) when we know that there is a defined rating temperature for white, red, rose and sparkling wine...or does it mean that the temperature of the room where the wines were tasted was 20 °C?

Author Response

The application of RATA method in wine analysis is not new but as the authors say never to such large and heterogeneous sample of wines. The main problem of this work is the presentation of achieved results.

We sincerely thank the reviewer for the time dedicated to evaluate our manuscript

Firstly explain was the list of modalities for odour/flavour one or there were two separate list, one for odour and one for flavour...

Thank you for your comment. There were two separate lists for odours and flavours. Lines 127-129 have been modified to make the concept clearer.

also in the figure there are marks for odour -O, flavour -F but also O/F...please explain how this was defined.

O/F in the figure legend refers to the fact that both odour and flavor from a specific attribute were removed from the analysis. This has been modified in the revised version of the manuscript.

All the figures with variables (figure b) have too much data and are not easy to read. My suggestion is to present the data in separate figures, one dealing with odours, one with flavours and the one with taste and tactile sensation. In this way also much better discussion of results can be performed.

Thank you very much for your comment, the authors are aware that the graphs are not easy to read due to the large number of descriptors. However, the authors feel that the inclusion of additional graphs would burden the manuscript.  The layout of the pictures has been modified in the hope that they will be more readable (vectors have been deleted).

Section 2.5 is written twice..

Thank you for your comment. The authors apologise for the mistake. This has been corrected.

Please explain why the wines were tasted at room temperature (20 °C) when we know that there is a defined rating temperature for white, red, rose and sparkling wine...or does it mean that the temperature of the room where the wines were tasted was 20 °C?

The authors acknowledge the fact that there are recommended temperatures for serving wine samples but in a laboratory context, even small differences in serving temperatures between samples can lead to large errors in sensory evaluation, therefore, all samples have been provided at the same temperature. This procedure is common practice and in line with other studies related to wine sensory evaluation:

Red wines:

-           Gonzaga et al 2021, DOI:10.20870/oeno-one.2021.55.4.4805

-           Gonzaga et al 2022 https://doi.org/10.1016/j.foodres.2021.110719

-           Hranilovic et al. 2021 https://doi.org/10.1016/j.foodchem.2021.129015

-           Mezei et al 2021, DOI:10.20870/oeno-one.2021.55.2.4571

White and sparkling wines:

-           Zhang et al 2022, https://doi.org/10.1016/j.foodchem.2022.133305

-           Carlin et al. 2022, https://doi.org/10.1016/j.foodres.2022.111404

-           Sancho-Galán et al 2022, https://doi.org/10.3390/foods11040509

-           White & Heymann (2015), doi: 10.5344/ajev.2014.14091

Round 2

Reviewer 1 Report

As far as i know, Italian wine has a  long history with more than 4,000 years. It's better for you to cite a correct article. 

Author Response

Dear reviewer,

thank you for your comment and suggestion. We have modified the sentence with "thousands of years of.." and cited an appropriate reference (Pomarici et al. 2021).

Reviewer 2 Report

No further coments..

Author Response

Dear reviewer,

thank you for the further revision of our manuscript. We are glad that the modifications made are satisfactory.

Regards

Monica Laureati on behalf of all authord